



# The influence of wind and land evapotranspiration on the variability of moisture sources and precipitation of the Yangtze River Valley

Astrid Fremme[1] and Harald Sodemann[1]

[1]Geophysical Institute, University of Bergen, and Bjerknes Centre for Climate Research, Bergen, Norway

**Correspondence:** Astrid Fremme (astrid.fremme@uib.no)

**Abstract.**

The Yangtze River Valley (YRV) experiences large intraseasonal and interannual precipitation variability, which is mainly due to East Asian monsoon influence. The East Asian monsoon is caused by interaction of many processes in the coupled land-atmosphere-ocean system. To better understand YRV precipitation variability in this complex system, we have studied the precipitation moisture sources and their connection to YRV precipitation. We obtained the moisture sources by using the ECMWF's ERA Interim reanalysis data set, the FLEXible PARTicle dispersion model (FLEXPART) and the WaterSip moisture source diagnostic. The variability of moisture sources reflects the variability of YRV precipitation. Intraseasonal variations of moisture sources include a shift of the most important source regions as the monsoon progresses. Interannual variability of the moisture sources shows that sources which are less important climatologically are closely connected to variations of the driest and wettest years. Our results show that land directly contributes 58% of moisture for YRV precipitation during 1980-2016, whereas the ocean contributes 42% in direct transport. While the importance of the ocean as a moisture source is often emphasized, our results underscore the importance of the process of continental recycling and the role of land moisture sources.

*Copyright statement.* TEXT

# 1 Introduction

The Yangtze River Valley (YRV) lies on the east coast of China. The region is affected by the East Asian monsoon and experiences dry winters and wet summers. Winters are dominated by cold and dry winds from continental regions to the north-west, while summer circulation is characterized by substantially more warm and moist southwesterly winds which bring the monsoon precipitation to the YRV (Huang et al., 2012). The focus of this study is on the wettest half of the year, namely, the months of April to September, in which the YRV receives 72% of its annual precipitation.

The population of the YRV in East China depends on monsoon rainfall for agriculture and water supply. At the same time, variability of monsoon rainfall can have negative consequences through droughts and floods (Huang et al., 2007; Piao et al., 2010). The mechanisms causing rainfall variability are not fully understood, and future changes in precipitation are uncertain





(Sun et al., 2016). A continued search for a better understanding of the processes causing rainfall variability therefore remains vital.

The East Asian monsoon precipitation exhibits variability regarding different aspects. The monsoon varies in its onset, rainfall amount, and spatial distribution (Ding and Chan, 2005). Being situated in a humid region, precipitation variability in the

YRV is mostly connected to flood episodes. In 1998, for example, regions along the Yangtze River experienced extraordinarily heavy floods, with on average $9.1\,\mathrm{mm\,day^{-1}}$ summer rainfall during that year (Jun and Chen, 2001). On the other hand, the driest summer since 1979 was the year of 2005, with an average of $4.4\,\mathrm{mm\,day^{-1}}$. Variability of the East Asian Monsoon is characterized by the interaction of many processes in the coupled land-atmosphere-ocean system. Some of these are the variability and strength of the monsoon circulation, the temperature of the surrounding oceans, the persistence of the Meiyu

front and the position of the Western Pacific Subtropical High (WPSH;  Ding and Chan, 2005; Feng et al., 2015).

The variability of moisture sources for a precipitation event can be affected by changes in factors such as evaporation and moisture transport or precipitation-causing mechanisms. The variability of moisture sources may be causing precipitation variability, but at the same time result from changes in other factors. In this study we seek to identify and understand factors contributing to intra-seasonal precipitation variability from such a moisture source perspective.

One of the prime factors that have been investigated as mechanisms behind precipitation variability is the variability of moisture contribution from the surrounding oceans. Since moisture origin is not a directly observable quantity, indirect, model-based methods have been used to determine the variation of this factor.

Studies with an emphasis on finding oceanic moisture sources have located and quantified the most important ocean regions as the Arabian Sea and the Bay of Bengal (BoB) as parts of the Northern Indian Ocean, the South China Sea (SCS), and the

East China Sea as part of the Western Pacific (Zhou and Yu, 2005; Wang and Chen, 2012; Chen et al., 2013; Zhou et al., 2010). Zhou and Yu (2005) for example, examine water vapor transport patterns corresponding to different positions of the main rain belt over East China. They find that the rain belt pattern associated with rainfall over the YRV receives moisture from midlatitude northeast water vapor as well as tropical southwest water vapor from BoB and SCS which can be traced back to the Philippine Sea. Zhou et al. (2010) find that the majority of moisture inflow to eastern China comes through the northern

boundary of the SCS with $200\times10^{6}\,\mathrm{kg\,s^{-1}}$ through this boundary. In their results an inflow of $152\times10^{6}\,\mathrm{kg\,s^{-1}}$ over a boundary at $100\,^{\circ}\mathrm{E}$ also suggests that land sources can play an important role for East China.

More recently the role of land regions as moisture sources to YRV has been more strongly recognized (Wei et al., 2012; Sun and Wang, 2015; Zhao et al., 2016; Rodríguez et al., 2017; Pan et al., 2017; Li et al., 2016; Drumond et al., 2011). Previous studies have used a range of different methods, and as a result, the location of key moisture sources vary between studies.

While some studies find land sources to be spread out over large regions (Wei et al., 2012; Sun and Wang, 2015; Zhao et al., 2016; Rodríguez et al., 2017; Pan et al., 2017), others emphasize the YRV region itself as the strongest land source (Li et al., 2016; Drumond et al., 2011). Wei et al. (2012) found that the most important moisture sources mainly lie in the pathways for moisture transport over land, and that the ocean plays an important role in initiating the transport. Local evapotranspiration in a region similar to the YRV accounts for about 10-15% during the wet season, while Indochina contributes 8-15%, South China

13-15%, Western Pacific 5-15%, SCS 6-12% and BoB 3-11%. Indochina, South China and the BoB are the most important





moisture sources during the precipitation peak of the monsoon. Pan et al. (2017) found that BoB and Arabian Sea contribute <4% during summer. They found the North Western Pacific to be the dominant oceanic source to YRV in other months (15.8-24.6%) than June and July (8.1-10.6%). In June and July the North Indian Ocean was the dominant oceanic source region with a contribution of 30%. The Indochina Peninsula contributed 9.9% of annual precipitation. Local YRV evaporation and South China had a combined contribution in summer exceeding 10%.

According to these latter studies, moisture contribution from the land surface provides an important fraction of the monsoon precipitation. The land surface provides moisture to the YRV through recycling of moisture from previous precipitation events. We use the term continental recycling for moisture recycling from any land region, while the term local recycling refers to recycling within the target region. During recycling events moisture originates from and interacts with the land surface. Disregarding moisture recycling can impact our view of moisture sources to YRV and their variability. Moisture recycling should be kept in mind when searching for possible mechanisms affecting the monsoon precipitation.

The lack of agreement with respect to both location and magnitude of the moisture sources for the YRV highlights the need for further attempts to locate the spatial distribution of moisture sources to the YRV, and their seasonal cycle. We apply here the state-of-the-art Lagrangian moisture source diagnostic of Sodemann et al. (2008), which provides a quantitative accounting of the contributions of evaporation along the flow path of air masses precipitating in a pre-defined target area.

Baker et al. (2015) used the method applied in this study for a 5-year period over a large region of China. In their study, the main focus was linking moisture source variations to the stable water isotope composition in cave deposits. Our present study, in contrast, examines the moisture sources of a more focused domain with relatively homogeneous precipitation regime. We first aim at finding robust features of the moisture sources distribution and intra-seasonal to inter-annual variability, covering a 37-year period. Next, we estimate the sources beyond the direct moisture contribution, exploring land and ocean moisture sources further back in time. Lastly, we explore local factors which might affect moisture sources and precipitation, before drawing our conclusions.

## 2   Method and Data

For this study we use a Lagrangian method to identify moisture sources to the Yangtze River Valley (YRV). Lagrangian methods follow the movement of air parcels through the atmosphere over time (Stohl et al., 1998; Stohl and Thomson, 1999). The humidity budget of an air parcel can be modified by evapotranspiration $e$ and precipitation $p$ as the airmass moves around the atmosphere (Stohl and James, 2004):

$$\Delta q = e - p \tag{1}$$

where $\Delta q$ is the change in specific humidity of an air parcel over a 6 h time period.

Different methods are in use to identify the moisture sources from trajectories. Here we use the Lagrangian moisture source diagnostic "WaterSip" (Sodemann et al., 2008). The WaterSip method assumes that for each 6 h time step either $e$ or $p$ will dominate while the other can be disregarded. Increases in specific humidity in the air parcels exceeding a threshold value $\Delta q_c$





are thus taken to be due to evaporation or transpiration from the surface, whereas decreases are due to precipitation. Data on evaporation and precipitation are not used directly, but rather estimated by $\Delta q$. Trajectories of air parcels precipitating over the target region are evaluated individually. Starting at 15 days before a precipitation event, at each time step, the fractional contribution of a humidity increase (thought to be due to dominating evaporation) to the previous specific humidity of the air

parcel at that time is calculated. In case of precipitation, previous evaporation regions are assumed to contribute according to the fraction they represent in the air parcel.

When part of the humidity of an air parcel precipitates, all earlier contributions contribute and are thereby discounted. For example, a mass of moisture originally gained by the air parcel ten days before reaching the target region might all be lost to precipitation the next day, still several days before reaching the target region. In this case, the earlier uptake and its region will

no longer be counted as a source for subsequent precipitation events by the air parcel.

This so-called moisture accounting provides a fractional contribution of each evaporation event to the final precipitation in the target area. Furthermore, it provides the percentage of the precipitation for which moisture sources have been identified. Notably, the method does not critically depend on the length of trajectories beyond a certain number of days. Due to the accounting method, expanding the analysis period from 10 days to 20 days, for example, typically only results in the identification

of an additional 5–10% of the moisture sources (Sodemann and Stohl, 2009). This particular property of the method contrasts with the widely used Lagrangian $e - p$ method (Stohl and James, 2004). There, the net effect of $e - p$ events for the air parcel trajectories are aggregated over a predefined time period, and results depend on the chosen aggregation period (e.g., Stohl et al., 2008).

We use the air parcel trajectory dataset of Läderach and Sodemann (2016) as a basis for the Lagrangian diagnostic WaterSip,

which has been extended by three years, and now covers the period 1980–2016. The dataset has been calculated using the Lagrangian particle dispersion model FLEXPART V8.2 (Stohl et al., 2005), using the 6-hourly European Centre for Medium-Range Weather Forecasts' ERA-Interim reanalysis (Dee et al., 2011). ERA-Interim (Dee et al., 2011) has been shown to be the best reanalysis datasets for representing monsoon precipitation (Lin et al., 2014; Huang et al., 2016). It is in good agreement with observations over monsoon regions, and specifically also over Eastern China (Liu et al., 2018). The trajectory

dataset of Läderach and Sodemann represents the global atmosphere by 5 million air parcels of equal mass. Trajectories contain the horizontal and vertical position and specific humidity along with other atmospheric variables (see Läderach and Sodemann, 2016, for further details). Trajectories are first extracted for all air parcels precipitating over the target region before the WaterSip method described above is applied.

The analysis region for the YRV spans 27°–33° N and 110°-122° E (Fig. 1, red box). We focus on the lower valley only, not

including to the upper reaches of the Yangtze River basin (west of 110° E), which experience a different precipitation regime (Chen et al., 2009).

Sources for precipitation over the ocean are excluded with a minimum threshold of 25 m elevation. Other thresholds for the moisture source diagnostic are 0.1 g kg$^{-1}$ for $\Delta q_c$, a trajectory length of 15 days, and relative humidity >80% for precipitation over YRV. No distinction is made for moisture uptake within and above the boundary layer. These thresholds lead to a good





representation of the spatial distribution and the seasonal cycle of precipitation over YRV (Fig. 2). The thresholds result in source attribution for 95% of the precipitation.

As part of this study, monthly ERA-Interim data for soil moisture, evaporation, and 850 hPa wind are used for direct comparison with moisture source results. For comparison with vegetation we include the Normalized Difference Vegetation Index (NDVI). For the NDVI we use the 1982–2015 monthly average of satellite-observed 3rd generation NDVI from National Oceanic and Atmospheric Administration's (NOAA) Advanced Very High Resolution Radiometer (AVHRR) (Pinzon and Tucker, 2014).

For validation, ERA Interim precipitation is compared to precipitation in the gridded observational dataset CN05.1 (Wu and Gao, 2013), which is based on observations in China. This dataset is used for the years 1980-2014.

## 3 Results and Discussion

### 3.1 Precipitation seasonality in East Asia

Precipitation over south and east Asia in the ERA Interim dataset has a maximum at the southern edge of the Tibetan Plateau and at the eastern border of the Bay of Bengal during April-September (Fig. 1). While large regions receive more than 10 mm day$^{-1}$ between April and September, the target region of this study, the lower reaches of the Yangtze river, denoted here as the Yangtze River Valley (YRV), receives on average 5.9 mm day$^{-1}$ of precipitation (Fig. 1, red box). YRV wet season precipitation shows a meridional gradient, with 4 mm day$^{-1}$ in the north and up to 8 mm day$^{-1}$ in the south (Fig. 2a). ERA Interim's representation of precipitation shows a similar spatial pattern as the high-resolution, gridded observational dataset CN05.1 (Wu and Gao, 2013)(Fig. 2a,b). The spatial pattern of precipitation obtained through the WaterSip method described in Sec. 2 is also similar (Fig. 2c).

The YRV has a pronounced precipitation seasonality. The six wettest months are April to September, and precipitation peaks in June (Fig. 2c). Both June and July are considered peak monsoon months (Ding and Chan, 2005). The monsoon precipitation and overall precipitation seasonality agrees well between the reanalysis, observations, and estimated from the WaterSip method (Fig. 2d), with on average 5.9, 5.9 and 5.2 mm day$^{-1}$ respectively. The similarity in both pattern, amount and seasonal cycle of precipitation validates the use of the WaterSip method for this region. Of the precipitation estimated by the WaterSip method, 95% is attributed to a source, while 5% is not accounted for, for example due to moisture sources further back in time than 15 days. The remainder of this study is only based on the moisture that can be attributed to a source.

### 3.2 Moisture sources of YRV precipitation

For each precipitation event in the YRV during the ERA-Interim period, we trace back to the moisture sources using the WaterSip method. The resulting average moisture sources for the YRV in April-May are shown in Fig. 3a. The shading can be interpreted as the contribution of evaporation to YRV precipitation in units of mm day$^{-1}$, equivalent to kg m$^{-2}$ day$^{-1}$. Moisture contributions range from 0 mm day$^{-1}$ in white to 1 mm day$^{-1}$ in black. The maximum contribution is from south-west China,



while large parts of Asia and the surrounding oceans contribute only small amounts. The 50th and 80th percentiles enclose 50% and 80% of the total moisture contribution by picking the grid points with largest contributions (Fig. 3a-c, red dashed lines). The extent of the percentiles denote the most important source regions. At the same time, the region between both red lines shows the importance of relatively moderate contributions spread out over a large area, contributing 30% of the moisture.

Through the course of the wet season, the most intense source region to the YRV gradually moves closer and more north-east as the monsoon progresses (Fig. 3a-c). At the start of the wet season, in April-May, the most pronounced source regions are over the western part of South China and the Indochina peninsula (Fig. 3a). In June-July the eastern part of south China becomes more important (Fig. 3b), and by August-September East China becomes the dominant moisture source to YRV (Fig. 3c). The largest source contributions for the two-month climatologies are during June-July, when the sum of the moisture source

contributions provide the YRV precipitation maximum. The mean precipitation in the YRV during June-July is $7.1\,\mathrm{mm\,day^{-1}}$ (Fig. 2d), with maxima of $8.8\,\mathrm{mm\,day^{-1}}$ in the south and west in the region (not shown). The overall maximum source is then over South China (Fig. 3b), with $0.97\,\mathrm{mm\,day^{-1}}$.

    The 80th percentiles show equally pronounced changes (Fig. 3a-c). In April-May the 80th percentiles cover mostly land regions, the South China Sea and small parts of the Western Pacific. In June-July the 80th percentile stretches out over India and

the Indian Ocean, while in August-September shifts further into the Western Pacific. The contribution from weaker moisture sources correspond to the area between the 50th and 80th percentiles. The region between the two percentiles is the source for 30% of the moisture, with a contribution below $0.2\,\mathrm{mm\,day^{-1}}$ in June-July.

    The two-month anomalies are shown with green (red) values for above (below) average contribution (Fig. 3d-e). The anomalies highlight the northward and eastward movement of the most important sources through the wet season. In April-May

(Fig. 3d) the southwestern edge of China and Myanmar contribute more than average, together with the South China Sea. In June-July (Fig. 3e) the Bay of Bengal, Indochina Peninsula and South China contribute more than average. At the end of the wet season, in August-September (Fig. 3f), Eastern China and the Yellow Sea are the only regions that contribute more than average. The reason for the high uptake within the YRV region in August-September is investigated further in Sec. 3.6. During August-September, the South China continental areas, the Indochina peninsula and the South China Sea all contribute less than

for the preceding wet months.

    Changes in the total contribution from all source regions are directly reflected in precipitation amount over YRV. Variability in moisture sources is thus intimately connected to variability of YRV precipitation. The YRV moisture sources show large seasonal variations, reflecting the large seasonal precipitation variations.

    The distribution of moisture sources for the YRV found here generally agrees with a range of previous studies (Wei et al.,

2012; Sun and Wang, 2015; Zhao et al., 2016; Rodríguez et al., 2017; Pan et al., 2017; Baker et al., 2015). However, our results are also in disagreement with studies focusing exclusively on oceanic moisture sources (Zhou and Yu, 2005; Wang and Chen, 2012; Chen et al., 2013; Zhou et al., 2010) or studies using the $e-p$ method (Li et al., 2016; Drumond et al., 2011). The results of this study are not necessarily in a direct contradiction to previous work, as there are different views on the effect which land recycling and the evaporation of moisture for precipitation has on the definition of moisture sources. Furthermore, each

method for studying moisture sources is associated with uncertainties. An advantage of the WaterSip method is that instead of





dealing with values for P and E obtained from model parametrization, these variables are estimated through more observation-restrained humidity changes in the atmosphere. As a first order result, the similarities to a range of previous studies using very different methods is encouraging. The results of this study are then further compared in more detail to the literature in Sec. 3.8.

### 3.3    Seasonal cycle of YRV moisture sources

The YRV moisture sources for April-September shows temporal changes, both with respect to location and amount. To examine the changes and the seasonal progression in more detail, we subdivide moisture sources into six land and four ocean regions (Fig. 4). The monthly climatology of the contribution from each region shows pronounced seasonality in all regions (Fig. 5).

In Fig. 5 the peak contribution of all regions ranges from 3.5 to $8 \times 10^{11}$ kg day$^{-1}$ (Fig. 5), which corresponds to 0.37 and 0.82 mm day$^{-1}$, respectively, if distributed evenly over the YRV region.

The spread of moisture contributions in Fig. 5 shows the interannual standard deviation of the moisture contributed from each source region. The regions with the largest standard deviation are the South China Sea (Fig. 5d) and the Western Pacific (Fig. 5e). The role of these and other sources regarding interannual variability and their connection to dry and wet years is explored further in Sec. 3.7.

Another feature of Fig. 5 is the timing of contribution from the different moisture sources. For spring and the pre-monsoon
(April-May) the South China Sea (Fig. 5d), South China (Fig. 5d), and the Myanmar region (Fig. 5b) are important moisture sources. They provide 16.2%, 17.3%, and 10.2% respectively (Table 1). Combined, this moisture contributes a substantial fraction of the pre-monsoon precipitation (43.7%).

During the monsoon precipitation peak months of June-July, contributions from the distant westernmost moisture sources of Bay of Bengal (Fig. 5b), India (Fig. 5a), and the Arabian Sea (Fig. 5a) show pronounced peaks. This coincides with the
strong westerlies of the Indian monsoon, and provides a link between the Indian and East Asian monsoons. These short-term, distant sources contribute 10.0%, 5.8% and 8.6% respectively in June-July (Table 1), a combined 24.4% of the total June-July moisture contribution. During June-July a peak in contribution can also be seen for the Indochina peninsula (Fig. 5c), and South China (Fig. 5d), although these regions also contribute substantially in spring. These two land regions contribute 11.0% and 14.9% in June-July (Table 1). The two land regions lie in the path of moisture arriving the YRV from the Indian ocean. We
hypothesize that moisture transported from the Indian Ocean, which precipitates and re-evaporates on its way to the YRV, will get the en-route land regions as their new moisture sources. This will be further investigated in Sec. 3.5.

For the late part of the monsoon, during August-September, the Western Pacific and the YRV region itself (Fig. 5e) become more important. During August-September these regions contribute 21.8% and 15.3% respectively (Table 1). This is a time when the region also experiences a decrease in moisture contribution from all other moisture source regions, suggesting a
changeover of moisture transport mechanisms in the monsoon system. This important transition period is further investigated in the next section (Sec. 3.4).

Regarding the causes of YRV precipitation variability, we note that different source regions are responsible for providing moisture for precipitation in the different stages of the monsoon. It is therefore conceivable that different mechanisms, such as continental recycling or long-range moisture transport, can play a role at different stages of the monsoon.





## 3.4 Continental recycling and regional evaporation recycling in the YRV

During the summer months, the YRV receives almost equal contributions from land and ocean sources. For the April-September wet season months, land sources contribute 57.8% of moisture for YRV precipitation while the ocean sources contribute 42.2% (Table 1). We refer here to contributions from land sources as *continental recycling* (Goessling and Reick, 2011). This term

has a wider perspective than *local recycling*, which only includes continental recycling from within the YRV (see Sec. 3.6). Continental recycling varies between 66% in February and 51% in August (Fig. 6). With more than half the moisture provided through continental recycling, this mechanism appears as important in sustaining YRV humidity and precipitation during the wet season.

Previous studies which considered land contributions found a range of values for continental recycling, from 40% Sun and

Wang (2015), a gradient of 30–50% for the YRV% Zhao et al. (2016), to a gradient of 30–60% over the YRV in summer Pan et al. (2017). As studies used different regions and time periods, continental recycling values are not directly comparable. The continental recycling fraction found in this study is nonetheless higher than what was found in previous studies. To further investigate the plausibility of this result, we compare moisture contribution from continental recycling to the total evapotranspiration (ET) due to the land surface and vegetation.

ERA-Interim mean ET over Asia in April-September shows a meridional gradient, with about 3-4 mm day⁻¹ in the south and below 1 mm day⁻¹ in the north (Fig. 7a). Ocean ET is stronger with maxima of over 5 mm day⁻¹ in the Bay of Bengal and Arabian Sea. ET values are generally lower, but the pattern resembles that of precipitation over the same regions (Fig. 2a).

While the ET displayed in Fig. 7a suggests a dominating role of the oceans, our analysis highlights that moisture source contributions to the YRV are only a subset of ET at the moisture source regions. The fraction $\epsilon$ of ET that ultimately arrives as

precipitation in the YRV is calculated as:

$$\epsilon = \frac{ET_{YRV,P}}{ET_{TOT}} \tag{2}$$

where $ET_{YRV,P}$ is the amount contributed from the moisture source to YRV precipitation, and $ET_{TOT}$ is the total ET. The $\epsilon$ within the YRV region is sometimes referred to as the regional evaporation recycling ratio (Van Der Ent et al., 2014), which shows the ratio of ET that subsequently precipitates within the same region.

For the April-September mean $\epsilon$ is below 30% (Fig. 7b) for all source regions. Values over South China are the highest, where more than 25% of ET in a small region results as YRV precipitation. The highest values over the ocean appear over the Western Pacific by the Yangtze River outlet and the South China Sea by the Indochina peninsula. The values of Fig. 7b underline that ET is clearly sufficient to fuel the moisture sources obtained in this study. This underlines the plausibility of the continental recycling values found in this study, and our moisture source results in general.

## 3.5 Second-order moisture sources of recycled precipitation

Moisture provided through continental recycling to YRV precipitation itself can have a local or remote origin. In this section we examine the sources of moisture from continental recycling arriving the YRV. To analyze the rate of which moisture is recycled



on land more than once, we first assume that continental moisture originates from precipitation within the same month. Then, continental moisture source regions are folded with the fraction of continental recycling in the source region.

We start with the continental moisture sources to YRV identified from the WaterSip method (Fig. 8a). Next, we calculate the local fraction of continental recycling for a region encompassing south, east and central Asia. In this calculation, monthly

averages were used to allow for a possible lag between precipitation and re-evaporation. The Asian continental recycling fraction shows a north to south gradient of decreasing continental recycling, coinciding with an increasing gradient with distance from the coast (Fig. 8b). Multiplication of the YRV moisture sources (Fig. 8a) and the Asian continental recycling fraction (Fig. 8b) yields the contribution of land sources to precipitation from continental sources to the YRV, termed here the second-order continental moisture sources (Fig. 8c). While Fig. 8c does not show the complete coverage of the second-order

moisture sources, it does provide information on the amount of YRV moisture which still has continental origin even before the last continental recycling event.

These results show that about two thirds (in summer) to three fourths (in winter) of the land source contributions to YRV have their origin over land, while one third (in summer) to one fourth (in winter) comes from the ocean. In combination with earlier results on the direct land and ocean contribution to YRV, this implies that the YRV has 42.2% of direct ocean contribution

for April-September precipitation, 17.0% continental recycling which is ocean contribution recycled once on land, while the remaining 40.8% of moisture to YRV has been recycled over land at least twice (Table 2).

South China and the Indochina peninsula are the two most important exterior continental moisture sources ot the YRV, contributing about 24% of summer precipitation. As already highlighted in Sec. 3.3, the June-July peak of the Indochina peninsula and South China moisture contribution might be connected to moisture transport from the Indian ocean, precipitating

and re-evaporating en-route to the YRV. The 50th and 80th percentiles of the moisture sources for that region extend over southeast Asia as well as over the surrounding oceans and India (Fig. 8c, dashed red lines). The sources for South China and the Indochina peninsula are therefore important second-order sources of the YRV.

The second-order continental sources show how moisture can be traced further back, sometimes back to when it evaporated from the ocean. Tracking moisture beyond the last place of evaporation is one of the reasons results of between previous

studies, but also between this study and others differ. The second-order sources emphasize the importance of the ocean in providing moisture which eventually undergoes continental recycling. The second-order sources also reveal the substantial fraction (40.8% in April-September) which is recycled on land more than once. Regarding the variability of the monsoon precipitation in the YRV, we note that the interaction with the land surface may therefore extend beyond the regions identified as first-order continental moisture sources.

## 3.6 Local recycling

Local recycling refers to the evaporation within a region contributing to precipitation within the region itself. Local recycling is therefore a subset of continental recycling (Fig. 6). For the YRV, the local recycling peaks in August, a time when contributions from all other sources except the Western Pacific have decreased compared to earlier months (Fig. 5e). Local recycling in the YRV is thus important for sustaining precipitation in the later part of the summer monsoon. YRV precipitation is lower in





August compared to June-July. Thereby, increased local recycling acted against a further decrease. The fractional contribution from local recycling increases from 9.8% in July to its highest of 15.8% in August (Fig. 9b). A peculiar finding is that local recycling peaks two months after the peak in contribution from moisture sources outside the target region (Fig. 9a, note the different scales). In this section, we investigate the possible reasons for the peak in local recycling in August by an analysis of

the seasonal evolution of characteristic variables of the YRV water cycle, including ET, soil moisture, NDVI, and local wind speed.

The ET rate within the region (Fig. 9c, blue) is high when local recycling peaks in August. However, ET peaks in July, when the fraction of local recycling is still quite low. The ET rate can be important for sustaining local recycling, but can not in itself explain why local recycling peaks in August. Figure 9c (green) shows the time evolution of $\epsilon$, similarly to (Fig. 7b). The

fraction of recycled ET is relatively stable throughout the year, except for the months of July, December and January. During nine months of the year, including August, approximately 12.7% of ET in the region returns as precipitation. In July, this part is reduced to 9.5%.

The soil moisture in the region (Fig. 9d, pink) also shows a seasonality distinct to local recycling. The gradual increase of soil moisture from January to July can not explain the abrupt increase in the local recycling fraction from July to August.

Since local recycling lags two months behind the precipitation peak, we explored the possibility that moisture from June or July precipitation was stored in the soil and affected August local recycling. However, interannual correlations of June or July soil moisture with local recycling in August are close to zero, both for absolute values of moisture contribution as well as the fraction of local recycling (not shown).

The NDVI is a satellite-observed index for the density of green leaves (Pinzon and Tucker, 2014). The NDVI average over

the region shows a gradual increase from January onward (Fig. 9d). NDVI peaks in August and stays high in September, similar to the local recycling fraction. This means vegetation and moisture released through transpiration could help support the local recycling peak in August and perseverance in September.

Finally, to compare local recycling with the circulation in the region, we used the 850 hPa mean wind speed over the YRV as an index (Fig. 9d). The wind speed over the region has a marked peak in July, concurrent with the decrease in recycled

ET (Fig. 9e). The stronger winds in July advect more moisture from distant sources, as well as a potentially stronger export of locally evaporated moisture. Contrarily, weaker winds can increase chances of locally evaporated moisture to re-precipitate within the region during August and subsequent months. At the time of the local recycling peak the region experiences some of the lowest wind speeds during the year, favoring higher local recycling rates.

In summary, the comparison between local recycling and characteristic variables of the water cycle in the YRV suggest that a

combination of factors is responsible for causing the late peak in local recycling, and maintaining late monsoon-season rainfall. Decreasing winds, high soil moisture, high green leaf area and strong solar forcing in combination lead to a sharp rise in local recycling and a slowed decline in rainfall seasonality in August. This suggests that rainfall variability in the late monsoon season is potentially affected by each of these factors, requiring a system-oriented approach to understanding variability of the YRV hydrological cycle.



### 3.7 Interannual variability of local recycling in summer

To explore the effects of local factors on the interannual variability of moisture sources, we now focus on the five driest and wettest summers out of the 37 summers between 1980–2016 (Table 3). Four of the five driest and wettest years in ERA Interim are matched by WaterSip as the most extreme. For all summers (JJA) the YRV average WaterSip precipitation estimate is

1.31 mm day$^{-1}$ lower than in ERA Interim. The WaterSip summer precipitation deviations range from -4% to -33%, with an average of -20.5%. This is a typical bias for Lagrangian diagnostics (Sodemann et al., 2008).

The total moisture supplied from all sources reflects the precipitation of the region, with anomalies of -22% and +28% during dry and wet years respectively. The relative changes in local variables are smaller (Table 3). YRV contribution is higher during wet than during dry summers. However, during wet summers the local recycling fraction is lower, suggesting that during wet

summers, the contribution from outside the region increases more than local contributions. Local ET, soil moisture and NDVI all change less than 5% in wet and dry summers compared to the mean. The 850 hPa wind speed over the region shows the largest changes, with 10% higher wind speeds for wet summers compared to the average, and 5% lower wind speeds during dry summers. The lower fraction of local recycling for wet summers and the high changes in wind speeds over the region suggest that outside contribution is more strongly connected to YRV precipitation variability than local moisture sources.

The small differences in local recycling between dry and wet summers motivates a comparison of moisture contribution from the different source regions for the five driest and wettest summers (Fig. 10). Contribution from all sources except the Western Pacific follow the same pattern, contributing more than average in wet summers and less than average in dry summers. The Western Pacific breaks the pattern and provides the least moisture for wet summers and less than average for dry summers. The changes between contribution in dry and wet summers are smallest for the YRV. This suggests that the YRV has a more stable

contribution to summer precipitation than sources outside the domain, and that contribution from the YRV does not intensify interannual variability. The largest changes in contribution between dry and wet summers are seen for South China and the Indochina Peninsula. South China provides -24% during dry summers and +32% during wet summers compared to its average summer contribution. The Indochina Peninsula provides -28% and +53% in dry and wet summers respectively. As these two land regions contribute a large fraction (24%) of summer precipitation moisture, their variability also plays a large role in the

interannual variability of YRV moisture sources and precipitation.

The South China Sea is the ocean region providing the largest amount of moisture in summer ($4.0 \times 10^{11}$ kg day$^{-1}$), and the Western Pacific provides the second largest amount ($3.7 \times 10^{11}$ kg day$^{-1}$). However, neither of these show the largest changes in contribution between dry and wet summers. The largest absolute changes between dry and wet years occur for the Indian Ocean sources of the Bay of Bengal (2.6 to $5.4 \times 10^{11}$ kg day$^{-1}$) and the Arabian sea (2.2 to $4.5 \times 10^{11}$ kg day$^{-1}$). The Indian

Ocean therefore seems to play a bigger role than other ocean regions for the interannual variability of YRV moisture sources and precipitation.

Guo et al. (2018) previously found that the major contributors of moisture influxes to different regions of China are not necessarily the major contributors to precipitation interannual variability. We arrive to a similar conclusion, although different study regions and methods hinder a direct comparison of our results. According to our findings, South China, the Indochina





peninsula and the Indian Ocean contribute the most to YRV summer precipitation interannual variability. On the other hand, the YRV region, the South China Sea and the Western Pacific are some of the major moisture sources, but contribute less to interannual variability.

### 3.8 Comparison with other studies

5 Based on the view of what constitutes a moisture source, previous studies on the moisture sources of the YRV can be divided into three groups. First, there are those that mainly consider ocean sources, which result in finding the most important ocean moisture sources (Zhou and Yu, 2005; Wang and Chen, 2012; Zhou et al., 2010). While knowledge of the ocean moisture sources is valuable, and one can argue that all moisture eventually comes from the ocean, it is first by including land sources in the analysis that we get the possibility to uncover the role of the land surface for moisture source variability.

10 The second group of studies estimated moisture sources as the net $e - p$ in the history of an air parcel (Chen et al., 2013; Li et al., 2016; Drumond et al., 2011). This view on moisture sources, while practical for finding net sources and sinks, has several drawbacks. The dominance of $p$ over $e$ in a 10-day integral map can mask the process of continental recycling, leading to underestimation of land sources. In addition, to prescribe equal significance to all moisture changes in a trajectory's history causes a high dependency on the choice of trajectory length. Results from the $e - p$ method will not show the last place of 15 evaporation for YRV precipitation.

Finally, there are a set of studies which, like this study, search for the regions where the moisture of a precipitation event last evaporated. A range of methods have been used, all with their separate advantages and disadvantages. The study of Wei et al. (2012) was based on the quasi-isentropic back-trajectory method (Dirmeyer and Brubaker, 2007), Sun and Wang (2015) used FLEXPART trajectories and an accounting method along trajectories similar to this study. The study of Zhao et al. (2016) 20 was based on the column water accounting method of van der Ent et al. (2010). Rodríguez et al. (2017) was based on a Met Office Unified Model climate simulation, and Pan et al. (2017), was based on a simulation with the climate model CAM5.1 and the MERRA reanalysis data. Without the ability to compare in detail, the results of these past studies are similar and do not contradict the results of this study. Although they are based on different data sources and examine slightly different regions, they all support that land is among the most important moisture source regions to the YRV and surrounding regions, with the 25 Indian Ocean providing an important part of the moisture for the monsoon precipitation peak, and large seasonal variations between contributions from different regions.

The method we have used in this study involving FLEXPART and WaterSip brings its own set of uncertainties. To be able to distinguish evaporation from precipitation events, the method assumes that either evaporation or precipitation dominates within each time step of 6 hours, and disregards the other (Sodemann et al., 2008). The choice of trajectory length can influence the 30 ability to find a source for a precipitation event (Sodemann and Stohl, 2009). A threshold for minimum moisture uptake and release from an air parcel is set to try to deal with numerical errors. This threshold also deals with the effect of air parcels mixing and thus introducing incorrect moisture sources. The threshold of minimum relative humidity for target region precipitation is the most influential threshold, and can affect the estimated precipitation over the target region if changed. Ultimately, the results are limited by the ability of ERA Interim to represent the actual state of the atmosphere.



When choosing a method to answer a research question, the definition of what constitutes a moisture source leads to large differences in results, and should be considered carefully. Still, for the methods with similar views on the definition of moisture sources, results are difficult to compare directly. It could be beneficial to have a common measure to compare the different results. For example, the mass-average moisture source distance for our results is $2433\pm376$ km during the summer months,

and the centroid of the moisture sources is located at $19°$ N and $100°$ E. Using these variables, future studies may be able to quantitatively compare their results to our present findings.

## 4    Conclusions

The Yangtze River Valley (YRV) is under the influence of the East Asian Monsoon, which causes dry winters and wet summers. In addition to large seasonal variations, the YRV also experiences large interannual variability. As a way to decipher the

underlying mechanisms for precipitation variability, we have studied the variability of YRV precipitation moisture sources using the ERA Interim reanalysis data set for the years 1980-2016. Trajectories from the Lagrangian model FLEXPART were used in combination with the the moisture source diagnostic tool "WaterSip" to quantify the moisture sources. Thereby, we take a perspective that allows for both continental and oceanic sources of moisture. The ocean was found to directly contribute 42% of moisture for precipitation in the YRV (Fig. 11). Furthermore, the ocean contributes moisture indirectly through the means

of continental recycling. Continental recycling allows the land surface to supply 58% of moisture for precipitation, where one third is ocean moisture recycled on land once before precipitating over the YRV, while two thirds are recycled on land more than once. According to our results, land moisture sources by means of continental recycling provide more than half of the precipitation in the YRV. Hence, factors at the land surface such as evapotranspiration, soil moisture and vegetation are likely to influence moisture source contributions.

The key results of this study are summarized below:

- Continental moisture sources supplied a large part (57.8%) of the moisture for the YRV precipitation. Although land contributions were large, the moisture supplied by land sources was well within the evapotranspiration rates at the source regions.

- Ocean moisture sources contributed 42.2% of moisture for YRV precipitation directly, and contributed more moisture
indirectly by means of continental recycling.

- Local recycling provides moisture for YRV precipitation from within the region. Local recycling peaks two months after the monsoon precipitation peak, and constitutes 15.8% in August.

- The intraseasonal variability of local recycling is related to a combination of the factors evapotranspiration, soil moisture, vegetation and 850 hPa wind speed in the YRV. Wind speed thereby appears as one of the main factors, and could likely
explain the late peak in local recycling.



- The second-order sources of the YRV precipitation consist of 17.6% ocean moisture which was recycled on land once, while 40.8% was recycled on land more than once. Results for second-order sources showed how land source regions receive moisture from a mix of land and ocean sources.

- Moisture sources for the five driest and wettest summers in the YRV are most closely connected to interannual variability of the ocean sources of the Bay of Bengal and the Arabian Sea, as well as the continental sources of the Indochina peninsula and South China. On the other hand, the South China Sea, the Western Pacific and the YRV region itself, while important for providing moisture in the climatology, are less important when it comes to interannual variability.

The results of this study support the view that land regions in a larger region of East Asia are critically important for moisture supply and precipitation variability of the YRV. This study also emphasizes that a different set of land and ocean moisture sources are important for sustaining the summer climatology, causing intraseasonal variability and interannual variability. This view serves as an important backdrop for understanding how land-atmosphere interactions influence YRV precipitation in past, present, and future climates.

*Code and data availability.* The code and data used in this study are available from the authors on request.

*Competing interests.* The authors declare no competing financial interests.

*Acknowledgements.* This work was supported by the project UTF-2016-long-term/10030. We acknowledge the Swiss National Science Foundation for partial funding through grant No. 200021_143436 "Spatial and Temporal Scales of Linkages in the Atmospheric Water Cycle (Waterscales)". This work used storage capacity from the Norwegian computing infrastructure NIRD through Project NS9054K (COPEWET). Access to the ECMWF ERA-Interim reanalysis data was provided through Met Norway.





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





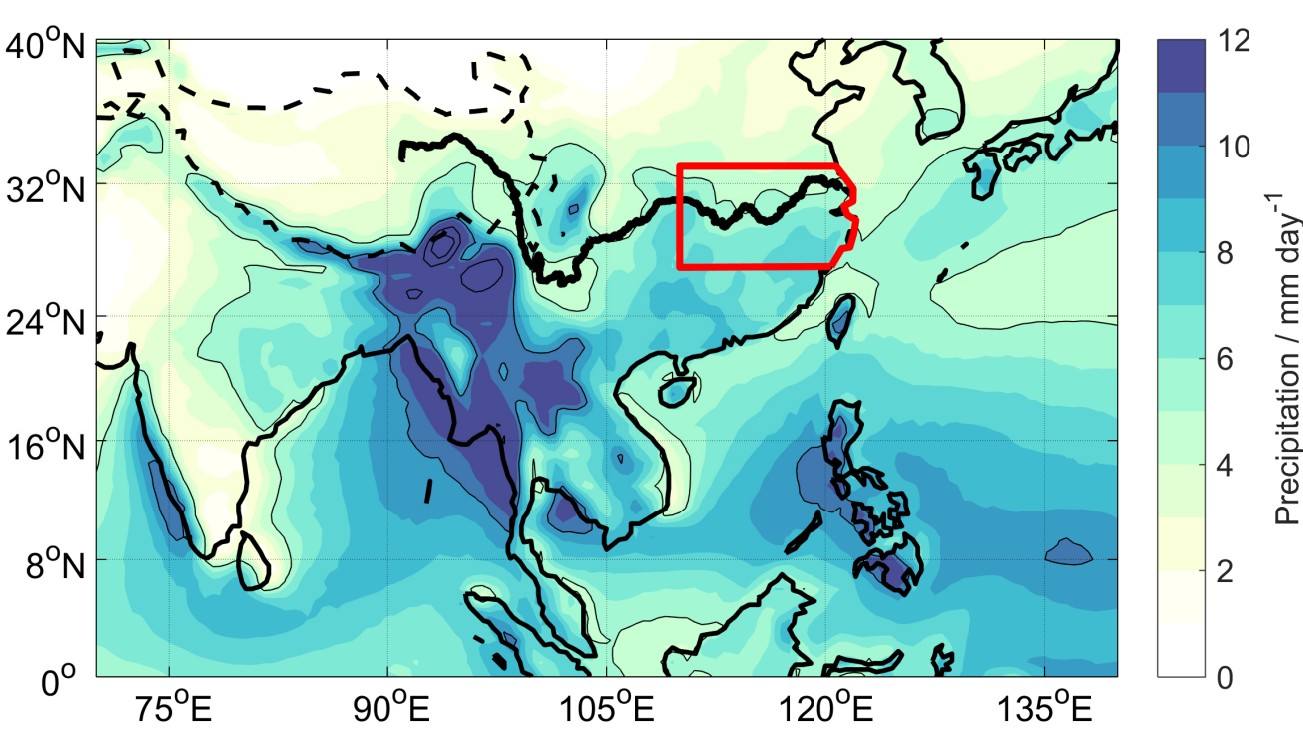

**Figure 1.** ERA Interim precipitation (shading) over south and east Asia in mm day$^{-1}$. Black ontours every 5 mm day$^{-1}$. The YRV target region (red), the Yangtze River (black), and the 4 km topography contour of the Tibetan Plateau (dotted line) are shown. Precipitation is the 1980-2016, April-September mean.





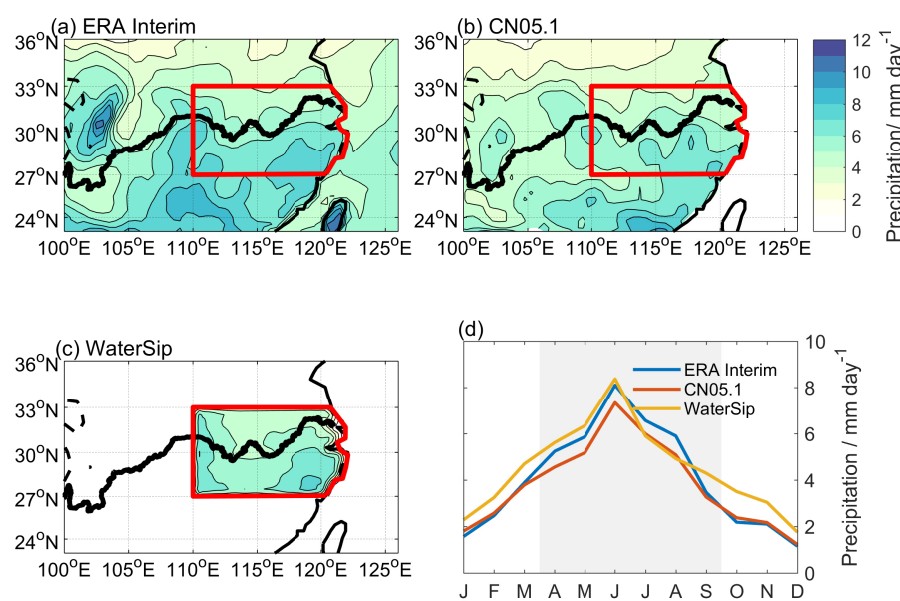

**Figure 2.** Precipitation over the YRV target region. April-September mean according to (a) ERA Interim, (b) the gridded observational dataset CN05.1, and (c) values estimated by the WaterSip method. The last panel (d) shows monthly precipitation over the YRV region for ERA Interim, CN05.1, and WaterSip. ERA Interim and WaterSip climatologies are for 1980-2016, while the CN05.1 climatology is for 1980-2014. Units are mm day$^{-1}$.



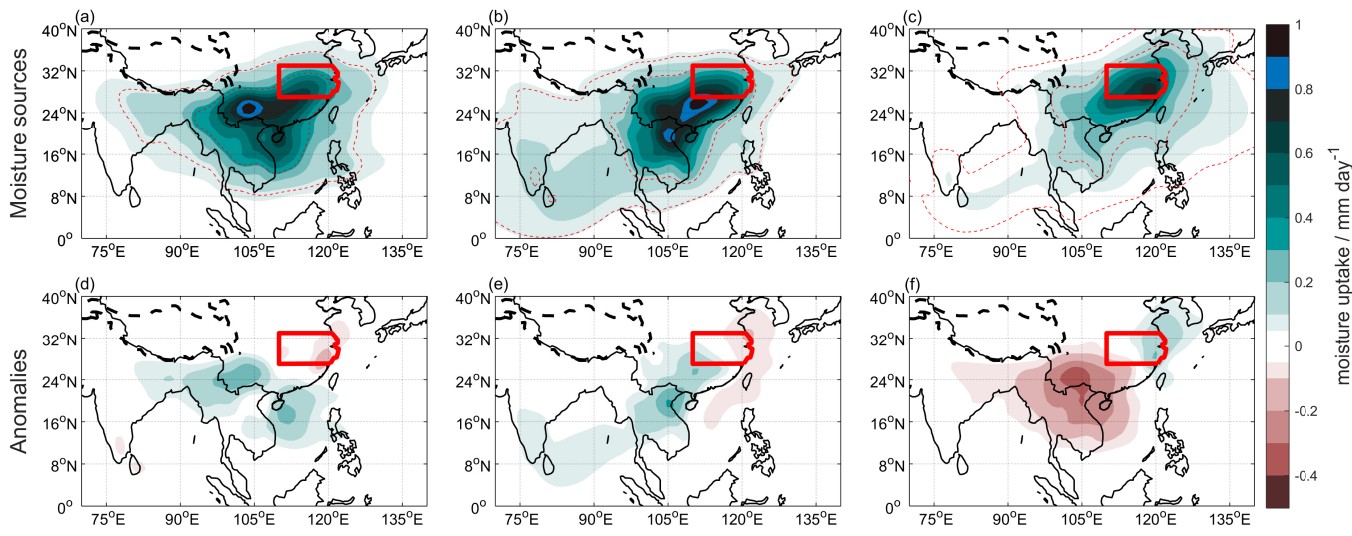

**Figure 3.** Two-month mean moisture sources for YRV for April-September. The upper panels (a-c) show the mean moisture sources for
(a) late spring (April-May), (b) mid summer (June-July), and (c) extended late summer (August-September). The lower panels show the (d)
April-May, (e) June-July and (f) August-September anomalies compared to the April-September mean. The target region (red) and 4000 m
elevation of the Tibetan Plateau (dotted contour) are shown. The 50th and 80th percentiles of the mass contributed from moisture sources are
shown in red dotted contours. The units are mm day$^{-1}$.





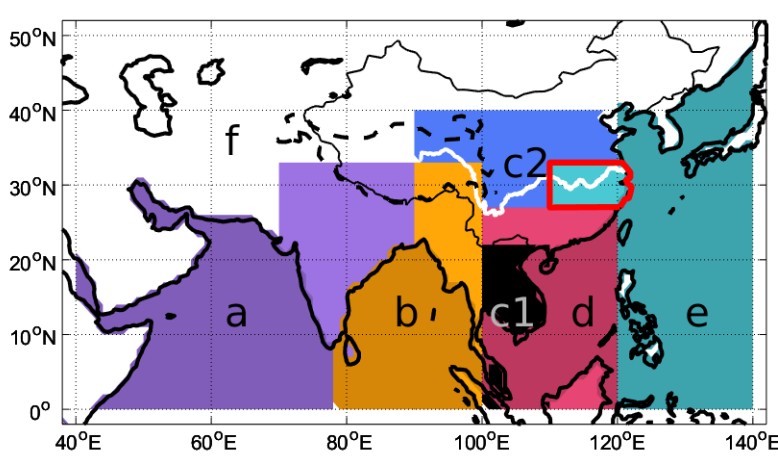

**Figure 4.** Definition of source regions. The different regions are named (a) Arabian Ocean and India, (b) Bay of Bengal and Myanmar, (c1) Indochina peninsula, (c2) Upper Yangtze River Valley, (d) South China Sea and South China, (e) Western Pacific and the target region of the Lower Yangtze River Valley, and (f) remaining ocean and land regions. The target region (red) and 4000 m elevation of the Tibetan Plateau (dotted contour) are shown.





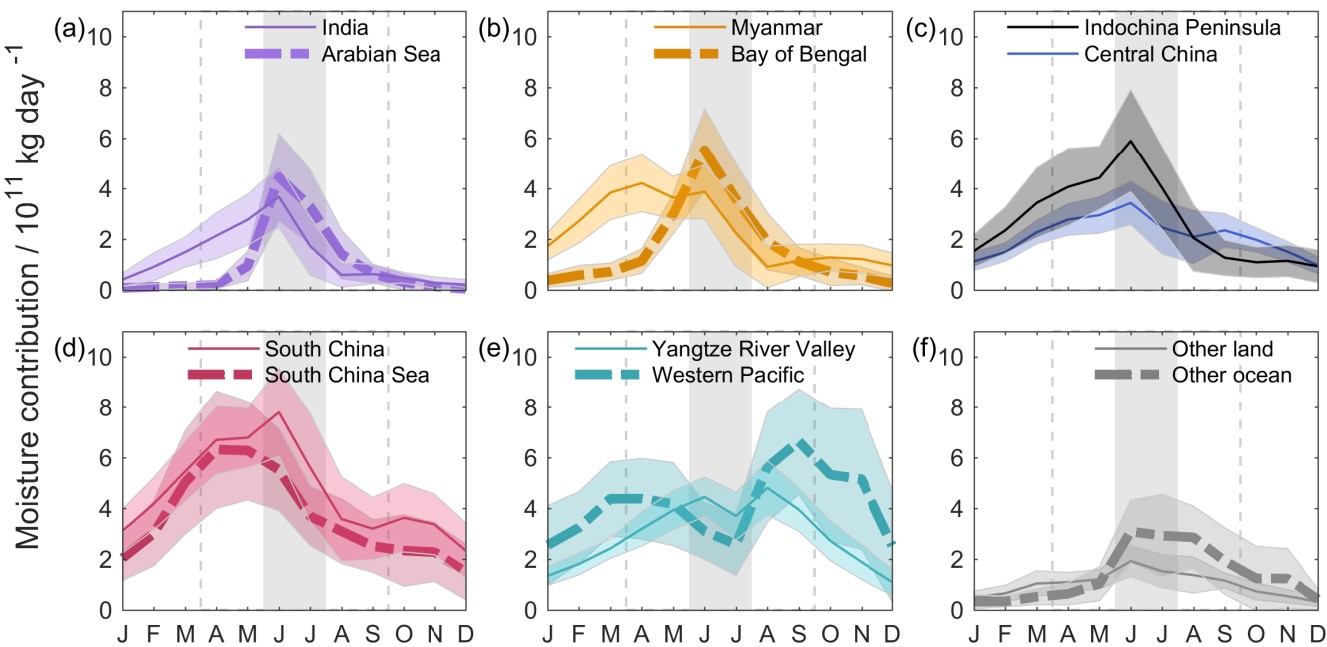

**Figure 5.** Moisture contribution from different source regions. Division and colors correspond to those in Fig. 4. Thick dashed lines show oceanic regions, while thin continuous lines show continental regions. The wettest half of the year is marked, with peak-precipitation-months June and July shaded. Contribution and standard deviation of contribution are shown for the different regions of: (a) Arabian Ocean and India, (b) Bay of Bengal and Myanmar, (c1) Indochina peninsula, (c2) Upper Yangtze River Valley, (d) South China Sea and South China, (e) Western Pacific and the target region of the Lower Yangtze River Valley, and (f) remaining ocean and land regions. The units are $10^{11}$ kg day$^{-1}$.



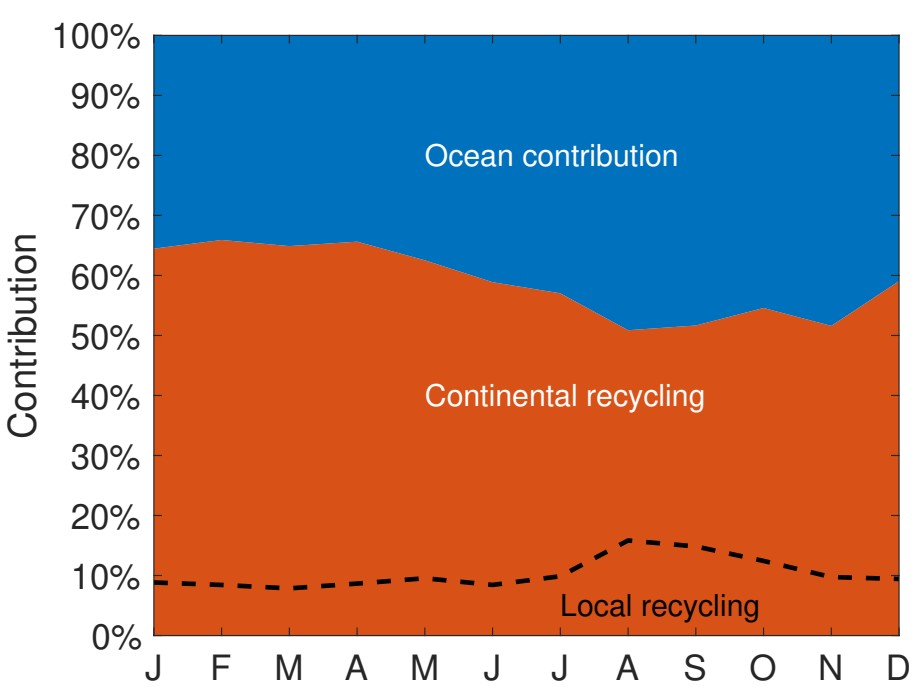

**Figure 6.** Fraction of land and ocean contribution to YRV precipitation. Local recycling is shown with a dashed line, and is a subset of continental recycling.





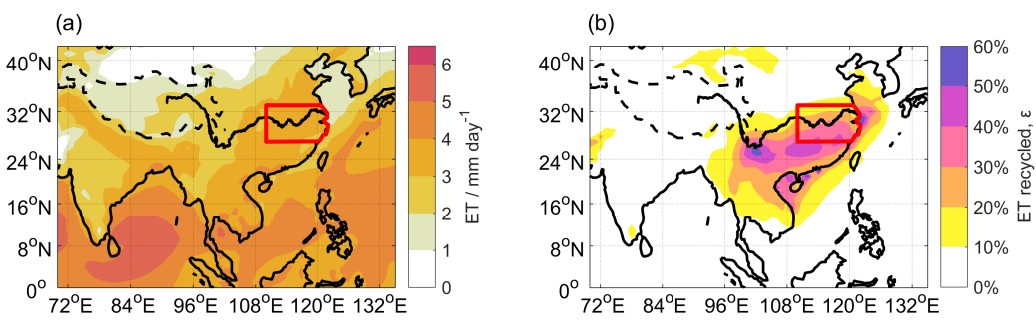

**Figure 7.** Evapotranspiration (a) and $\epsilon$, the fraction of evapotranspiration resulting as YRV precipitation (b). The values are the April-September, 1980-2016 means. ET data is from ERA Interim and in $\mathrm{mm\,day^{-1}}$, while $\epsilon$ is a combination of WaterSip results and ET.





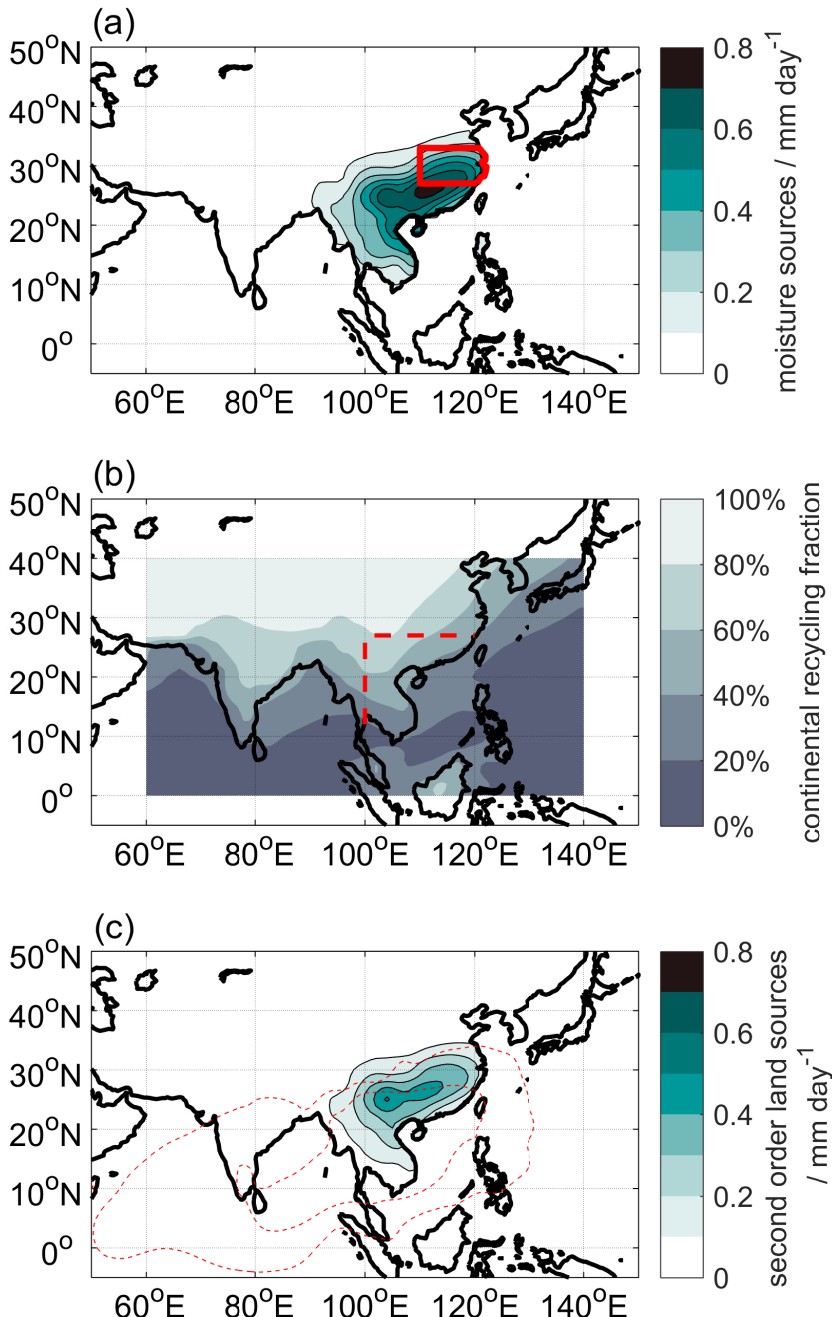

**Figure 8.** Identification of second order land sources. Continental moisture sources to YRV are shown in mm day$^{-1}$ (a), with the YRV as a red box and the South China and Indochina peninsula regions demarcated as one by red dashed lines. (b) shows the fraction of continental recycling to a larger section of Asia. (c) shows in shading the second order land contribution in mm day$^{-1}$, which is the product of (a) and (b), and in red dashed lines the 50th and 80th percentiles of moisture sources to South China and the Indochina peninsula.





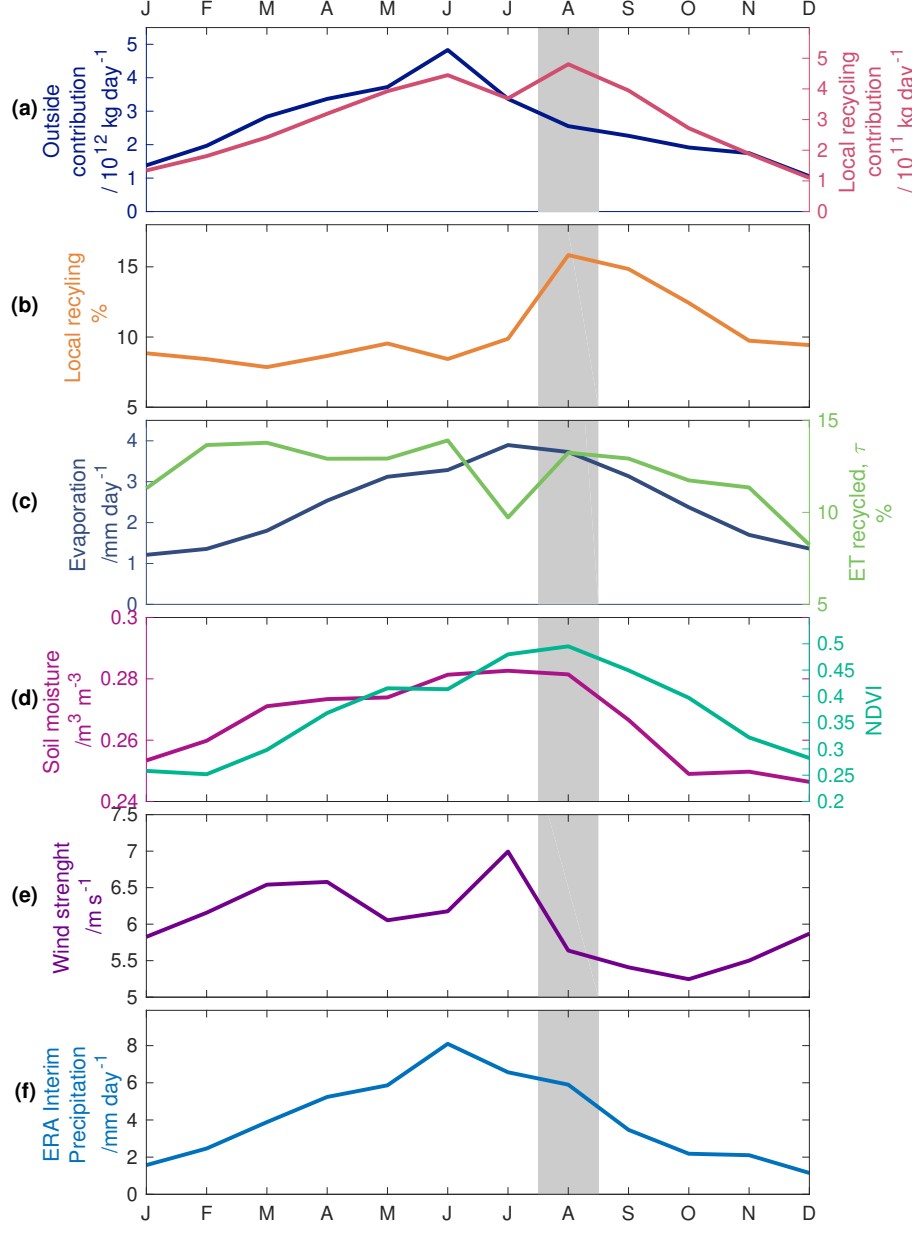

**Figure 9.** Seasonal cycle of YRV variables. The panels show (a) absolute values of moisture contribution from regions outside YRV, and within YRV in $10^{12}$ kg day$^{-1}$ and $10^{11}$ kg day$^{-1}$ respectively, (b) local recycling in percent, (c) ERA Interim evapotranspiration over YRV in mm day$^{-1}$, and $\epsilon$, the percentage of evapotranspiration over YRV which is recycled, (d) ERA Interim soil moisture in m$^3$ m$^{-3}$ and NDVI (unitless), (e) ERA Interim 850 hPa wind strength over the region in m s$^{-1}$, and (f) ERA Interim precipitation in mm day$^{-1}$. All values are monthly climatologies for 1980-2016 except NDVI which is the 1982-2015 climatology. August is shaded, showing the month of the local recycling peak.





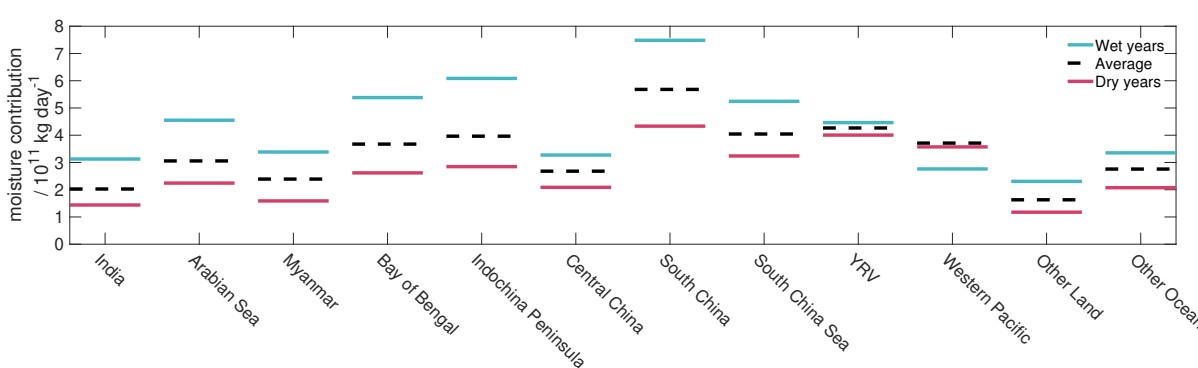

**Figure 10.** Moisture contribution during the five driest (1981, 1985, 2003, 2013) and wettest (1993, 1995, 1996, 1998, 1999) summers. The average contribution for all summers is also shown with dashed lines. The extent of the source regions are defined in Fig. 4.





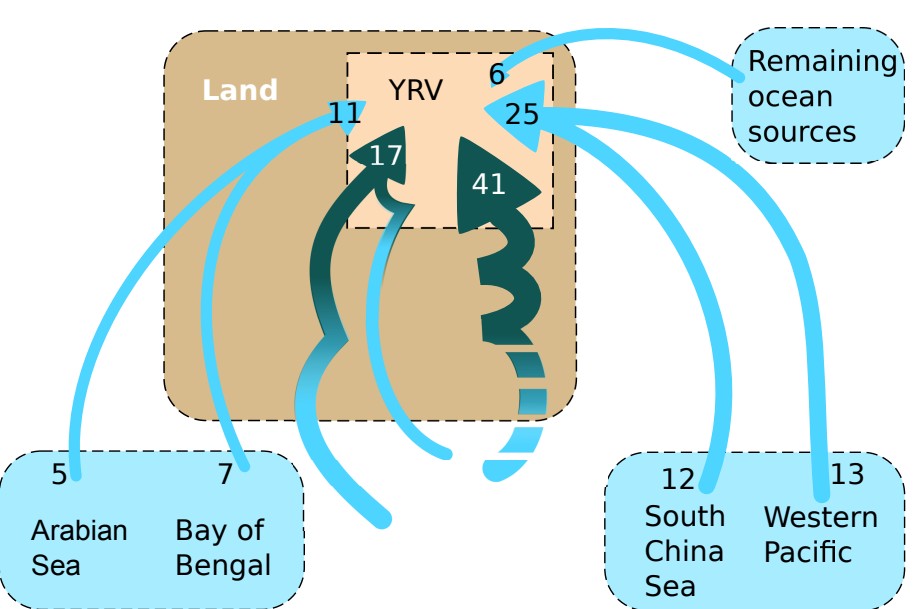

**Figure 11.** Moisture sources of the YRV during April-September. All numbers are contributions expressed as the percentage of YRV precipitation. Light blue arrows represent direct ocean contribution, while light-to-dark arrows represent oceanic moisture with unknown origin recycled once (left), or more than once (right) before precipitating over the YRV.





**Table 1.** Moisture source contribution fraction to the Yangtze River Valley (YRV)

|  | April-May | June-July | August-September | Wet season mean |
|---|---|---|---|---|
| *Land sources* |  |  |  |  |
| India | 6.3% | 5.8% | 2.2% | 4.8% |
| Myanmar | 10.2% | 6.7% | 3.7% | 6.9% |
| Indochina Peninsula | 10.9% | 11.0% | 5.8% | 9.2% |
| Central China | 7.4% | 6.6% | 7.9% | 7.3% |
| South China | 17.3% | 14.9% | 11.9% | 14.7% |
| Yangtze River Valley | 9.1% | 9.2% | 15.3% | 11.2% |
| Remaining land regions | 2.3% | 3.9% | 4.4% | 3.8% |
| *Land sources total* | 64.1% | 58.0% | 51.3% | **57.8%** |
| *Ocean sources* |  |  |  |  |
| Arabian Sea | 1.4% | 8.6% | 3.6% | 4.5% |
| Bay of Bengal | 5.2% | 10.0% | 5.3% | 6.8% |
| South China Sea | 16.2% | 10.2% | 9.8% | 12.1% |
| Western Pacific | 11.0% | 6.4% | 21.8% | 13.1% |
| Remaining ocean sources | 2.1% | 6.8% | 8.3% | 5.8% |
| *Ocean Sources total* | 35.9% | 42.0% | 48.7% | **42.2%** |





**Table 2.** Second order moisture sources to YRV

|  | Second order land contribution / $10^{12}$ kg day$^{-1}$ | % of YRV precipitation recycled more than once | % of YRV precipitation recycled once only | % of YRV precipitation directly from ocean sources |
|---|---|---|---|---|
| April-May | 1.87 | 48.0% | 16.1% | 35.9% |
| June-July | 1.70 | 37.4% | 20.5% | 42.1% |
| August-September | 1.02 | 36.0% | 15.3% | 48.7% |
| Wet season mean | 1.53 | 40.8% | 17.0% | 42.2% |
| All year | 1.23 | 42.7% | 16.2% | 41.1% |





**Table 3.** Local variables for the five driest (1981, 1985, 2003, 2013) and wettest (1993, 1995, 1996, 1998, 1999) YRV summers.

|  | Driest 5 |  | Average | Wettest 5 |  |
| --- | --- | --- | --- | --- | --- |
| Moisture supply / Precipitation ($\times 10^{12}\,\mathrm{kg\,day^{-1}}$) | 3.12 | -22.2% | 4.02 | 5.14 | +28.0% |
| YRV contribution ($\times 10^{11}\,\mathrm{kg\,day^{-1}}$) | 4.00 | -6.12% | 4.27 | 4.46 | +4.62% |
| Local recycling | 13% |  | 11% | 9% |  |
| ET ($\mathrm{mm\,day^{-1}}$) | 3.8 | +4% | 3.6 | 3.5 | -5% |
| Soil moisture ($\mathrm{m^3\,m^{-3}}$) | 0.2714 | -3.6% | 0.2818 | 0.2950 | +4.7% |
| NDVI* | 0.47 | +2.6% | 0.46 | 0.46 | -1.2% |
| Wind speed ($\mathrm{m\,s^{-1}}$) | 5.8 | -4.8% | 6.1 | 6.7 | +10.2% |

* NDVI data only included for years between 1982-2015.