# Peer review of "The influence of wind and land evapotranspiration on the variability of moisture sources and precipitation of the Yangtze River Valley"

_Hydrology and Earth System Sciences, 2018_

## Referee Comment (RC1) · Anonymous Referee #1 · 11 Mar 2019

**Astrid Fremme and Harald Sodemann**

**Anonymous Referee #1**

General comments:

The moisture source for precipitation over the lower Yangtze River Valley has been studied in this manuscript. The study is conducted using trajectories calculated from a Lagrangain particle dispersion model "FLEXPART V8.2" and a Lagrangian moisture source diagnostic "WaterSip". This is an important topic with profound socio-economic values and has been widely studied previously. Authors of this manuscript make effort to compare findings herein with previous studies and explore new diagnoses to further evaluate contributions from land and ocean, i.e., the second-order moisture source

from land. Although, it is just briefly mentioned in the manuscript, the idea of finding a common measure to compare different results from different studies is encouraging. I would like suggest to accept this manuscript if authors sufficiently answer following questions.

Specific comments:

> Page 5, Lines 24-26 and Page 11, Lines 4-6: information inconsistency. In former, I have learned that 95% of moisture source is attributable by WaterSip to precipitation. However, in latter, it says, I quote, "The WaterSip summer precipitation deviations . . . with an average of -20.5%. This is a typical bias for Lagrangian diagnostics (Sodemann et al., 2008)." Which one the actual WaterSip accuracy in terms of estimating precipitation?

> Page 6, Line 18: In the sentence, "the two-month anomalies" is mentioned without giving the referring mean. By reading the caption of Figure 3, I learn that this is the anomaly against the whole wet season (April-September). However, it is confusing without an explicit mention in the text.

> Section 3.5: I cannot find information on how is the local fraction of continental recycling calculated (and in Figure 8b). Without this information, I cannon justify the validity of the second-order continental moisture source, and therefore, the whole section 3.5. In my opinion, the innovation of this study largely comes from this section.

> Page 13, Lines 4-5: I do not know how is the mass-average moisture source distance defined. Is it defined from the furthest boundary of moisture source to the center of the YRV along the great circle? What does the deviation stand for, monthly variation or interannual variation? And, why there is not deviation for the centroid of moisture sources?

Technical corrections:

> Page 8, Lines 9-11: please put citations in the correct parentheses.

> Page 9, Lines 24-25: this sentence is incomprehensible.

> Figure 8. The red dashed lines is shown in the wrong panel; or the caption is wrong.

---

## Referee Comment (RC2) · Anonymous Referee #2 · 12 Mar 2019

This study used a Lagrangian model FLEXPART to analyze the moisture sources of Yangtze River Valley (YRV) during summer based on the "Watersip" method and the ERA-Interim reanalysis data. The dataset and methods used in this study are reliable. Although there have been several studies focusing on the moisture sources of precipitation in the YRV region, this study provided detailed discussion on the seasonal cycle, continental and local recycling, intraseaonal and interannual variability of the moisture sources of YRV region, which makes this study a valuable study. The results of this study show a consistency with the previous studies regarding the importance of continental moisture source for the precipitation in the YRV region, and also show some new findings regarding the second-order moisture source and the interannual variability of moisture sources. On the other hand, there are some issues in this paper needing to be addressed. I suggest that the authors should address these questions before this paper is published. Specific questions: (1) The authors divided the Section 3 into eight subdivisions, which makes the key points in the results not highlighted. The readers may what is the focus of this study when reading through these eight subdivisions. I suggest the authors to highlight the key points in Section 3, where the number of sub-divisions in Section 3 may be needed. For instance, if the focuses of this study are the contientnal recycling the interannual variability, the main body of Section 3 should be associated with these two issues. The subsection 3.1 "precipitation seasonality" is a background knowledge, which could be combined with the subsection 3.2 "moisture sources of YRV precipitation". In addition, the title of this paper is "the influence of wind and land evapotranspiration. . .". However, only section 3.6 and 3.7 gave a discussion on the influence of wind, while the other six subdivisions in Section 3 did not mention wind at all. It makes the reader wonder whether the wind speed is a key factor in this study. (2) In section 3.7, the authors concluded that the Indian Ocean play an impor-tant role for the interannual variability of YRV moisture sources and precipitation, and the South China Sea and Western Pacific contribute less to the interannual variabil-ity. According to Fig. 10, the moisture contribution changes from 3.1 to 5.1x10ˆ11 kg dayˆ-1 between dry and wet summers for South China Sea. This change is just slightly smaller than the change of moisture contrition for the Arabian Sea, which suggests that the South China Sea is also an important moisture source for the interannual variability. In addition, in Fig. 10, the pattern of South China Sea is distinct from the pattern of Western Pacific. It is not reasonable to put the two source regions into the same cate-gory. (3) I tried to understand Fig. 8 and the discussion on Fig. 8, but it seems difficult to understand the information in Fig. 8b and 8c. I suggest to clarify what is "the fraction of continental recycling to a larger section of Asia" (Fig. 8b). (4) "Sources for precipi-tation over the ocean are excluded with a minimum threshold of 25m elevation." What

does this mean? (5) "Other thresholds for ….and relative humidty>80% for precipitation over YRV". Does this mean that only the air parcels with a relative humidity>80% were traced back? Why not trace back all the air parcels that have a release of moisture within the YRV region? (6) "for the YRV% Zhao et al. (2016)". I think the "YRV%" is a typo. (7) In the end of section 3.6, the authors concluded that "Decreasing winds…and strong solar forcing in combination lead to a sharp rise in local recycling…". But there is no discussion on the influence of solar forcing in the previous discussion. (8) In section 4, the first term of the key results, "Although land contributions were large, the moisture supplied by land sources was well within the evapotranspiration rates at the source regions." I don't quite understand the meaning of this sentence. The land source regions contribute moisture to atmosphere via evapotranspiration. I think this is a well-understood process. Why the authors said "Although land contributions were large, … was well within the evapotranspiration rates…" ? (9) In section 4, the fifth term of the key results, "…17.6% was recycled on land once, 40.8% was recycled on land more than once." 17.6%+40.8% = 58.4%. In the first term of the key results, it is mentioned that the continental moisture sources supplied 57.8% of the moisture for the YRV precipitation. 58.4% and 57.8% are not consistent, although they are two close numbers.

---

## Referee Comment (RC3) · Anonymous Referee #3 · 15 Mar 2019

General comments The authors investigated the moisture sources of the Yangtze River Valley in terms of the land and ocean contributions. It is useful for YRV humidity and precipitation prediction when showing more than half the moisture provided to RYV precipitation through continental recycling, this mechanism. Generally speaking, the paper is well written and documented, the discussion and comparison with precious studies are sufficient, and tables and graphics are well constructed. The few questions and comments I have are listed below in the specific comments to the authors. Specific comments 1. As from the title of this paper, I know that this work focuses on or is

about "the influence of wind and land evapotranspiration on the variability of moisture sources and precipitation of the Yangtze River Valley", but I didn't get any conclusion or statement about the influence of wind and land evapotranspiration on the variability of moisture sources in the abstract. Maybe the 58% contribution of land directly involves the land evapotranspiration, the wind was not mentioned, at least. So I suggest the authors rewritten the abstract or revise the title. 2. Page 4, Line 19-20, is the air parcel trajectory dataset of Läderach and Sodemann available online? If it is, it is better to give the accessible link here. 3. Page 5, Line 21, Fig.2c ——>Fig. 2d. Please check it. 4. Page 10, Line 1-4, it is better to exchange the order of the Figure 9a and 9b. 5. Page 10, Line 24, Fig.9d ——>Fig. 9e. Please check it. 6. Page 10, Line 31, I didn't find any variable involving the "strong solar forcing". 7. Page 27, Line 31, there are only four driest years showed here (1981, 1985, 2003, 2013). Please check it.

---

## Referee Comment (RC4) · Anonymous Referee #4 · 27 Mar 2019

General comments

The manuscript has analyzed the climatology, intraseasonal and interannual variability of moisture sources for the rainfall in Yangtze River Valley (YRV). The continental moisture sources are found to contribute more moisture for the rainfall in YRV than oceanic moisture sources if only the direct contribution is considered. The study have also found that the peak of the moisture sources from YRV is two months later than that of the moisture sources from the regions outside YRV due to the late peak in NDVI and lower-tropospheric horizontal wind over YRV. Besides, the study implies that the

key moisture sources are different between intraseasonal and interannual variability of the rainfall in YRV. The results are of significance in the understanding of the mechanism for the rainfall variability in YRV. However, several concerns should be addressed before the manuscript can be suitable for published.

Specific comments

1.The introduction puts forward the main scientific problem of the present research at Line 12-15 of Page 3. However, the main assignment of this study, which is outlined in Line 13-15 of Page 3, does not clearly provide sufficient information on how to resolve the problem. Moreover, I do not find the direct answer to problem throughout the manuscript. The sentence "Without the ability to compare in detail, the results of these past studies are similar and do not contradict the results of this study" may make the readers think that the problem are not really solved in this study. I suggest the authors to improve the proposal of the scientific problems.

2.The manuscript have studied too many issues regarding the moisture sources for the rainfall in YRV, which are too dispersed for the readers to understand the central idea of the study. So I suggest the authors to reorganize the results and discussion section (Section 3) to make it more concentrated.

3.Line 23-29 of Page 9. The paragraph gives the reasons for the disagreement among existing studies from the perspective of the way of considering second-order continental sources. However, no more detail is provided here. I suggest the authors to give the ways how existing studies track the moisture beyond the last place of evaporation, which may provide the evidence supporting the assertion.

Minors

1.Line 32 of Page 4. Why sources for precipitation over the ocean are excluded by including the sources above 25 m elevation, not zero m? The area of the land regions with elevation <25 m is actually not small.

[Figure]

2.Line 9-11 of Page 8. The sentence is hard to understand.

3.Line 15-16 of Page 8. Fig. 7a only shows the results in the southeast part of Asia, how can the ET results over the whole Asia be seen?

4.The title of the manuscript does not exactly match the results of the present study and is thus misleading. The title suggests that the manuscript aims to study the effect of surface wind velocity and land evapotranspiration variations on the variations of the moisture sources for the rainfall in YRV. This is actually only one of the issues of the manuscript (Section 3.6). So the title should be revised according to the scientific problem of the manuscript.

5.Line 31 of Page 10. This sentence emphasizes the role of soil moisture and solar forcing in causing the late peak of local recycling in August. However, this sentence contradicts with the results shown in Line 13-18 of Page 10. Also, I do not find any evidence for the solar forcing throughout the manuscript.

6.Line 5-6 of Page 11. The meaning of the sentence is hard to understand because what the precipitation deviations are is not provided. Also I do not find the corresponding supporting data in Table 3.

---

## Author Comment (AC1) · 1 Apr 2019

We would like to thank anonymous referee 1 with their considerate comments and help in making the manuscript more understandable. Following is a list of how the specific comments have been addressed:

*Page 5, Lines 24-26 and Page 11, Lines 4-6: information inconsistency. In former, I have learned that 95% of moisture source is attributable by WaterSip to precipitation. However, in latter, it says, I quote, "The WaterSip summer precipitation deviations . .*

[Figure]

*. with an average of -20.5%. This is a typical bias for Lagrangian diagnostics (Sodemann et al., 2008)." Which one the actual WaterSip accuracy in terms of estimating precipitation?*

Concerning the accuracy of WaterSip in terms of estimating precipitation, we clarify our statements to show that there is no inconsistency. P5, L24-26 (P6, L2-5) in the revised version read: "While there is an overestimation during most months of the year, WaterSip underestimates ERA Interim precipitation in summer (JJA) with an average of 20.5%. Of the precipitation estimated by the WaterSip method, 95% is attributed to a source, while 5% is not accounted for, for example due to moisture sources further back in time than 15 days."

*P6, L18: In the sentence, "the two-month anomalies" is mentioned without giving the referring mean. By reading the caption of Figure 3, I learn that this is the anomaly against the whole wet season (April-September). However, it is confusing without an explicit mention in the text.*

Page 6, Line 18 (now Page 6, Line 30): How "the two-month anomalies" is calculated is now mentioned in the text. "Comparing each two-month period to the overall wet season mean we obtain the two-month anomalies (Fig. 3d-e)."

*Section 3.5: I cannot find information on how is the local fraction of continental recycling calculated (and in Figure 8b). Without this information, I cannon justify the validity of the second-order continental moisture source, and therefore, the whole section 3.5. In my opinion, the innovation of this study largely comes from this section.*

Section 3.5 (now 4.4): Information on how the local fraction of continental recycling has been added in the revised manuscript. A new paragraph in the method section (Page 4, Lines 20-29) now reads: "The WaterSip diagnostic tool is also used to obtain the so-called second-order moisture sources. This measure gives us more information on the number of times moisture goes through precipitation and re-evaporation over land before reaching the target region. Obtaining the second-order moisture sources

is a three step process in addition to obtaining the YRV moisture sources. Firstly, the moisture sources to a larger region of Asia are calculated, and the land fraction to the Asian region is obtained. This land contribution fraction is found by analyzing each trajectory separately. Knowing the moisture sources and relative contribution to each precipitation event, the land fraction is calculated. Secondly, the monthly mean land fraction over the Asian region is obtained by weighting by the contribution from each trajectory to precipitation over the region. For the third step we assume that continental moisture originates from precipitation in the same region within the same month. Folding the YRV land moisture sources by the fraction of land contribution to the source regions then gives the second-order moisture source land fraction to the YRV."

*Page 13, Lines 4-5: I do not know how is the mass-average moisture source distance defined. Is it defined from the furthest boundary of moisture source to the center of the YRV along the great circle? What does the deviation stand for, monthly variation or interannual variation? And, why there is not deviation for the centroid of moisture sources?*

Page 13, Lines 4-5 (now Page 13, Lines 22-26): Definitions for the mass-average moisture source distance and the centroid of the moisture sources are now stated, as well as their monthly standard deviations. "For example, the summer mass-average moisture source distance for our results is 2420 km with a monthly standard deviation of $\pm$376 km. The mass-average moisture source distance describes the distance between all moisture source evaporation events and the corresponding target region precipitation events, weighted by their contribution to precipitation in the target area. This is equivalent to the distance between the centroid of the moisture sources. The centroid of the moisture sources in out results is located at 19° N and 100° E."

*Page 8, Lines 9-11: please put citations in the correct parentheses.* Parentheses of citations were corrected.

[Figure]

*Page 9, Lines 24-25: this sentence is incomprehensible.* Page 9, Lines 24-25: The sentence "Tracking moisture beyond the last place of evaporation is one of the reasons results of between previous studies, but also between this study and others differ." has been changed to (Page 10, Lines 2-3): "An advantage of the approach used here is the ability to quantify the degree to which moisture undergoes multiple recycling events (see Sec. 2)."

*Figure 8. The red dashed lines is shown in the wrong panel; or the caption is wrong.*

Red dashed lines were changed to the correct panel (Fig. 8a).

———————————————

---

## Author Comment (AC2) · 1 Apr 2019

We would like to thank the referee for their helpful comments and suggestions, which we have implement as detailed below.

*(1) The authors divided the Section 3 into eight subdivisions, which makes the key points in the results not highlighted. The readers may what is the focus of this study when reading through these eight subdivisions. I suggest the authors to highlight the key points in Section 3, where the number of sub-divisions in Section 3 may be needed.*

*For instance, if the focuses of this study are the continental recycling the interannual variability, the main body of Section 3 should be associated with these two issues. The subsection 3.1 "precipitation seasonality" is a background knowledge, which could be combined with the subsection 3.2 "moisture sources of YRV precipitation". In addition, the title of this paper is "the influence of wind and land evapotranspiration. . .". However, only section 3.6 and 3.7 gave a discussion on the influence of wind, while the other six subdivisions in Section 3 did not mention wind at all. It makes the reader wonder whether the wind speed is a key factor in this study.*

- As suggested by the reviewer, the title of the manuscript has been changed to "The role of land and ocean evaporation on the variability of precipitation in the Yangtze River Valley" to better reflect the overall contents. The sections of the manuscript have been rearranged, with two revised section titles (3. Data and method validation, and 5. Discussion). This limits the number of subsections in the Results section, and provides an overall more logical structure. In addition, some of the subsections have been given new titles to better reflect what we want to convey through each of them. The subsections under Results are now: 4.1 Climatological mean moisture sources of YRV precipitation, 4.2 Mean seasonal cycle of YRV moisture sources, 4.3 Continental recycling and regional evaporation recycling in the YRV, 4.4 Second-order moisture sources of recycled precipitation, 4.5 Factors governing local recycling, and 4.6 Interannual variability of local recycling and distant contribution in summer. The total number of subsections has been kept the same, as we think they provide the best way to make our findings accessible to the readers.

*(2) In section 3.7, the authors concluded that the Indian Ocean play an important role for the interannual variability of YRV moisture sources and precipitation, and the South China Sea and Western Pacific contribute less to the interannual variability. According to Fig. 10, the moisture contribution changes from 3.1 to $5.1x10^{11}kgday^{-1}$ between dry and wet summers for South China Sea. This change is just slightly smaller than the change of moisture contrition for the Arabian Sea, which suggests that the South China*

*Sea is also an important moisture source for the interannual variability. In addition, in Fig. 10, the pattern of South China Sea is distinct from the pattern of Western Pacific. It is not reasonable to put the two source regions into the same category.*

- This is an interesting observation. The role of the South China Sea variability, which is slightly lower than for each of the Indian Ocean sources is now included in former section 3.7. Page 12, line 12-13:" The South China Sea is next (3.2 to 5.2 $\times 10^{11} kg day^{-1}$). The Indian Ocean sources therefore seem to play the largest role for the interannual variability of YRV moisture sources and precipitation, with the South China Sea following slightly behind."

*(3) I tried to understand Fig. 8 and the discussion on Fig. 8, but it seems difficult to understand the information in Fig. 8b and 8c. I suggest to clarify what is "the fraction of continental recycling to a larger section of Asia" (Fig. 8b).*

- A clarification of "the fraction of continental recycling" has been added. A new paragraph in the method section (Page 4, Lines 20-29) now reads: "The WaterSip diagnostic tool is also used to obtain the so-called second-order moisture sources. This measure gives us more information on the number of times moisture goes through precipitation and re-evaporation over land before reaching the target region. Obtaining the second-order moisture sources is a three step process in addition to obtaining the YRV moisture sources. Firstly, the moisture sources to a larger region of Asia are calculated, and the land fraction to the Asian region is obtained. This land contribution fraction is found by analyzing each trajectory separately. Knowing the moisture sources and relative contribution to each precipitation event, the land fraction is calculated. Secondly, the monthly mean land fraction over the Asian region is obtained by weighting by the contribution from each trajectory to precipitation over the region. For the third step we assume that continental moisture originates from precipitation in the same region within the same month. Folding the YRV land moisture sources by the fraction of land contribution to the source regions then gives the second-order moisture source land fraction to the YRV."

[Figure]

*(4) "Sources for precipitation over the ocean are excluded with a minimum threshold of 25m elevation." What does this mean?*

- The sentence concerning the minimum threshold of 25m elevation is changed to: (now Page 5, line 8):" The target region is limited to land areas with a threshold of 25m minimum elevation". What we mean by this is that oceanic regions within the 27°–33° N and 110°-122° E definition are excluded as part of the target region.

*(5) "Other thresholds for . . ..and relative humidty >80% for precipitation over YRV". Does this mean that only the air parcels with a relative humidity>80% were traced back? Why not trace back all the air parcels that have a release of moisture within the YRV region?*

- The relative humidity threshold is necessary for the WaterSip method to provide meaningful results. Without a threshold in relative humidity for precipitation over the target region, the precipitation estimate found using changes in specific humidity in the trajectories over the target region (as described in the methods section) will be heavily over-estimated by including humidity changes that are due to interpolation errors. Sensitivity tests have shown that a threshold of >80% gives reasonable results for the YRV.

*(6) "for the YRV% Zhao et al. (2016)". I think the "YRV%" is a typo.*

- corrected.

*(7) In the end of section 3.6, the authors concluded that "Decreasing winds. . .and strong solar forcing in combination lead to a sharp rise in local recycling. . .". But there is no discussion on the influence of solar forcing in the previous discussion.*

- The reference to "strong solar forcing" has been changed to "high evaporation rates", and the sentence (Page 11, line 13-14) now reads: "Decreasing winds, high soil moisture, high green leaf area and high evaporation rates in combination lead to a sharp rise in local recycling and a slowed decline in rainfall seasonality in August."

*(8) In section 4, the first term of the key results, "Although land contributions were large, the moisture supplied by land sources was well within the evapotranspiration rates at the source regions." I don't quite understand the meaning of this sentence. The land source regions contribute moisture to atmosphere via evapotranspiration. I think this is a well-understood process. Why the authors said "Although land contributions were large, . . . was well within the evapotranspiration rates. . ." ?*

- The first term of the key results in the conclusions has been rephrased (Page 14, Line 9-11): "Continental moisture sources supplied a large part (58.4%) of the moisture for the YRV precipitation. At first sight this number might seem high. However, comparing with reanalysis evapotranspiration rates at the source regions we showed that results were in a reasonable range." For readers more accustomed to methods focusing on oceanic moisture sources, the high land contribution might be in contrast to expectations.

*(9) In section 4, the fifth term of the key results, ". . .17.6% was recycled on land once, 40.8% was recycled on land more than once." 17.6%+40.8% = 58.4%. In the first term of the key results, it is mentioned that the continental moisture sources supplied 57.8% of the moisture for the YRV precipitation. 58.4% and 57.8% are not consistent, although they are two close numbers.*

- The referee is right in that the percentages of land contribution did not exactly add up. This was due to differences in weighting the averages. A consistent way of weighting the averages are now used across the whole manuscript, and the method of averaging was stated as (Page 30, Table 1) "Averages are weighted by monthly contribution". This changes the last decimal of many of the percentages given in the manuscript. The new values can be seen in the updated Table 1.

---

## Author Comment (AC3) · 1 Apr 2019

The authors would like to thank Anonymous Referee 3 for their considerate and useful comments. Following is a response to each of the posed comments:

*1. As from the title of this paper, I know that this work focuses on or is about "the influence of wind and land evapotranspiration on the variability of moisture sources and precipitation of the Yangtze River Valley", but I didn't get any conclusion or statement about the influence of wind and land evapotranspiration on the variability of moisture*

[Figure]

*sources in the abstract. Maybe the 58% contribution of land directly involves the land evapotranspiration, the wind was not mentioned, at least. So I suggest the authors rewritten the abstract or revise the title.*

- The title of the manuscript has been changed to "The role of land and ocean evaporation on the variability of precipitation in the Yangtze River Valley" to better reflect the contents.

*2. Page 4, Line 19-20, is the air parcel trajectory dataset of Läderach and Sodemann available online? If it is, it is better to give the accessible link here.*

- The air parcel trajectory dataset of Läderach and Sodemann is not available online, and therefore no link has been included.

*3. Page 5, Line 21, Fig.2c - 2d. Please check it.*

- The reference to the Fig. 2d was wrong as is now corrected.

*4. Page 10, Line 1-4, it is better to exchange the order of the Figure 9a and Fig. 9b.*

- The order of panel a and b in Figure 9 were exchanged as suggested.

*5. Page 10, Line 24, Fig.9d - 9e. Please check it.*

- The reference to the Fig. 9e was wrong as is now corrected.

*6. Page 10, Line 31, I didn't find any variable involving the "strong solar forcing".*

- The reference to "strong solar forcing" has been changed to "high evaporation rates", and the sentence now reads (Page 11, Line 13-14): "Decreasing winds, high soil moisture, high green leaf area and high evaporation rates in combination lead to a sharp rise in local recycling and a slowed decline in rainfall seasonality in August."

*7. Page 27, Line 31, there are only four driest years showed here (1981, 1985, 2003, 2013). Please check it.*

- The year 2006 was missing as a dry year in the captions, and has been inserted.

---

## Author Comment (AC4) · 1 Apr 2019

We would like to thank the anonymous referee 4 with their considerate comments and help in making the manuscript more understandable. Following is a list of how the comments have been addressed:

*Specific comments*

*1.The introduction puts forward the main scientific problem of the present research at Line 12-15 of Page 3. However, the main assignment of this study, which is outlined*

*in Line 13-15 of Page 3, does not clearly provide sufficient information on how to re-solve the problem. Moreover, I do not find the direct answer to problem throughout the manuscript. The sentence "Without the ability to compare in detail, the results of these past studies are similar and do not contradict the results of this study" may make the readers think that the problem are not really solved in this study. I suggest the authors to improve the proposal of the scientific problems.*

Page 3, Line 13-15 has been rewritten:"The lack of agreement with respect to both location and magnitude of the moisture sources for the YRV highlights the need for further attempts to locate the spatial distribution of moisture sources to the YRV, the moisture contributions from land and ocean, and the seasonal cycle of the moisture sources. " We have also made our comparison with other studies clearer. Page 13, Line 8-10: "Based on the location of the moisture sources and the seasonal cycle, the study of Rodriguez et al. (2017) and that of Pan et al. (2017) showed the most similarities to our results. As this study used a very different method to these, we conclude that these results are the most reliable."

*2.The manuscript have studied too many issues regarding the moisture sources for the rainfall in YRV, which are too dispersed for the readers to understand the central idea of the study. So I suggest the authors to reorganize the results and discussion section (Section 3) to make it more concentrated.*

As stated in the response to referee 2, the sections of the manuscript have been rear-ranged, with two revised section titles (3. Data and method validation, and 5. Discus-sion). This limits the number of subsections in the Results section, and provides an overall more logical structure. In addition, some of the subsections have been given new titles to better reflect what we want to convey through each of them. The sub-sections under Results are now: 4.1 Climatological mean moisture sources of YRV precipitation, 4.2 Mean seasonal cycle of YRV moisture sources, 4.3 Continental recy-cling and regional evaporation recycling in the YRV, 4.4 Second-order moisture sources of recycled precipitation, 4.5 Factors governing local recycling, and 4.6 Interannual

variability of local recycling and distant contribution in summer. The total number of subsections has been kept the same, as we think they provide the best way to make our findings accessible to the readers.

*3.Line 23-29 of Page 9. The paragraph gives the reasons for the disagreement among existing studies from the perspective of the way of considering second-order continental sources. However, no more detail is provided here. I suggest the authors to give the ways how existing studies track the moisture beyond the last place of evaporation, which may provide the evidence supporting the assertion.*

These lines have been rewritten: "An advantage of the approach used here is the ability to quantify the degree to which moisture undergoes multiple recycling events (see Sec. 2)." In Section 2 some methodological aspects are addressed.

*Minors*

*1.Line 32 of Page 4. Why sources for precipitation over the ocean are excluded by including the sources above 25 m elevation, not zero m? The area of the land regions with elevation <25 m is actually not small.*

This sentence was unclear and has been rewritten: "The target region is limited to land areas with a threshold of 25 m minimum elevation. " The 25m threshold is used to delineate the YRV target region. Here, the difference between using 0 and 25m is small (0.6% of the region). For moisture sources over land and ocean a land mask is used.

*2.Line 9-11 of Page 8. The sentence is hard to understand.*

The sentence has been rewritten: "Previous studies which considered land contributions to YRV precipitation reported between 30% and 60% continental recycling for different seasons and slightly different target regions, with a gradient of lower continental recycling to the southeast in the region, and more land contributions to the northwest (Sun and want, 2015; Zhao et al., 2016; Pan et al.,2017)."

*3.Line 15-16 of Page 8. Fig. 7a only shows the results in the southeast part of Asia, how can the ET results over the whole Asia be seen?*

- The phrase "Asia" has been changed to (Page 8, Line 28): "South and East Asia".

*4.The title of the manuscript does not exactly match the results of the present study and is thus misleading. The title suggests that the manuscript aims to study the effect of surface wind velocity and land evapotranspiration variations on the variations of the moisture sources for the rainfall in YRV. This is actually only one of the issues of the manuscript (Section 3.6). So the title should be revised according to the scientific problem of the manuscript.*

- As suggested by the reviewers, the title of the manuscript has been changed to "The role of land and ocean evaporation on the variability of precipitation in the Yangtze River Valley" to better reflect the overall contents.

*5.Line 31 of Page 10. This sentence emphasizes the role of soil moisture and solar forcing in causing the late peak of local recycling in August. However, this sentence contradicts with the results shown in Line 13-18 of Page 10. Also, I do not find any evidence for the solar forcing throughout the manuscript.*

- "A sentence has been added to the first results on soil moisture. (Page 10,Line 32)" We recognize that the soil moisture may participate in causing the late peak in local recycling, but is not a driving factor." - (Page 11, Line 13): "Solar forcing" has been changed to "evaporation rates".

*6.Line 5-6 of Page 11. The meaning of the sentence is hard to understand because what the precipitation deviations are is not provided. Also I do not find the corresponding supporting data in Table 3.*

- Part of this information is now given together with Fig. 2d, where the WaterSip underestimation of precipitation during summer can be seen. Page 6, Line2-3: "While there is an overestimation during most months of the year, WaterSip underestimates ERA

Interim precipitation in summer (JJA) with an average of 20.5%".